# Review: Botulinum Toxin for Treatment of Focal Limb Dystonia

**DOI:** 10.3390/toxins17030122

**Published:** 2025-03-04

**Authors:** Emma H. Kaplan, Michele Vecchio, David M. Simpson

**Affiliations:** 1Department of Neurology, Mount Sinai Hospital, New York, NY 10029, USA; 2Department of Biomedical and Biotechnological Sciences, University of Catania, 95123 Catania, Italy; michele.vecchio@unict.it; 3Physical Medicine and Rehabilitation Unit, A.O.U. Policlinico G. Rodolico S. Marco, 95123 Catania, Italy; 4Department of Neurology, Icahn School of Medicine at Mount Sinai, New York, NY 10029, USA; avid.simpson@mssm.edu

**Keywords:** focal dystonia, task-specific dystonia, botulinum toxin (BoNT)

## Abstract

Focal limb dystonias (FLDs) are abnormal postures and muscle contractions in an arm or leg that can occur in the setting of specific activities or without any stimulus. This pathology can have a profound impact on quality of life and potentially limit work in those whose dystonias are brought on by activities related to their occupations. Botulinum toxin (BoNT) is approved for use in the United States by the Food and Drug Administration for several indications, including cervical dystonia and blepharospasm, but not for FLD. Despite this limitation, BoNT is frequently used clinically for FLD, generally with individualized dosing based on patient need and clinician expertise. Various methods exist for targeting treatment to the specific affected muscles and assessing the benefit of treatment. Small clinical trials have demonstrated the efficacy of BoNT, but larger controlled studies are needed.

## 1. Defining Focal Limb Dystonias

While generalized dystonias may cause involuntary muscle contractions and abnormal postures in many muscles throughout the body, focal dystonias are defined as those affecting one single body part. Many focal dystonias are task-specific, meaning that they are brought out by performing a specific action, which would be a single arm or leg in the case of focal limb dystonia (FLD).

Task-specific dystonias can affect any number of body parts, including the mouth, neck, and larynx, but when occurring in the limbs, are one of the most common forms of FLD. Given the variety of specialized and unique tasks performed by the upper limb, those task-specific dystonias are more common than in the lower extremity, particularly in adults [1]. Some of the best known task-specific dystonias in the upper extremity occur with writing in the case of writer’s or stenographer’s cramp, or playing an instrument such as piano or drums in musician’s dystonia [2,3].

Embouchure dystonia is another form of musician’s dystonia related to facial movement with woodwind instruments, and does not fall under FLD [2]. Other reported cases of task-specific dystonia are associated with occupations and hobbies. In sports, they are known colloquially as “the yips,” particularly in golfers, though also noted in a variety of other sports, including pistol-shooting, tennis, and basketball, among others [4]. Although less common, lower limb task-specific dystonia may occur with ambulation, and, in some cases, are specific to running or stair climbing. Runner’s dystonia in particular may spread or generalize [1,5].

Non-task-specific dystonias can also occur in the limbs, either as a progressive course in those initially presenting with task-specific dystonia or as secondary forms of dystonia that may occur with stroke, trauma, parkinsonism, and other causes [1,6]. There may be overlap in the features of dystonia and spasticity in patients with central nervous system processes, such as stroke, or other causes of acquired brain injury [7,8].

## 2. Botulinum Toxin (BoNT) in the Context of Dystonia Treatment

Dystonias can have a profound impact on quality of life, including impairment of function in daily life and occupations, which may lead to difficulties with work or hobbies, and may cause social embarrassment and physical pain. Treatments for dystonia are aimed at reducing unwanted contractions and movements in an attempt to help the patient reduce these negative effects.

Treatment can include a variety of oral, surgical, rehabilitative, and local treatments, though some are better studied in generalized dystonias or are used in dystonias associated with other symptoms and conditions, such as spasticity [9]. Oral medications that have been tried include anticholinergics, baclofen, benzodiazepines, and medications targeting dopaminergic neurotransmission. Surgical options include intrathecal baclofen and deep brain stimulation, while selective denervation as a permanent surgical procedure is used primarily for cervical dystonia [10,11]. Physical and occupational therapy may also be helpful, particularly in task-specific dystonia. Small studies have shown some success, either with splinting the dystonic fingers (constraint-induced movement therapy) or, alternatively, with splinting the dystonic limb for a rest period prior to retraining [9].

BoNT has become a mainstay of treatment, particularly in focal dystonia, which is targeted to affected areas to reduce the systemic effects that may be seen with oral therapies. It is often used in combination with some of the other therapies listed above.

## 3. Physiology

Focal hand dystonia is associated with the loss of inhibition and dysfunction that may occur on a cortical and muscular level, such as abnormal sensorimotor cortical firing and impairment of the reciprocal inhibition of antagonist muscles that normally acts via afferent nerves from agonist muscles [12]. Although many dystonias with a known genetic basis are generalized, there may be gene variants associated with task-specific dystonias, such as in arylsulfatase G in musician’s dystonia and writer’s cramp, with less association present in blepharospasm in one study [13,14,15].

The classic understanding of the mechanism of BoNT is its weakening of muscles by breaking the connection between presynaptic nerves and muscle fibers across the neuromuscular junction. BoNT, in its active form, is made up of one heavy and one light chain protein connected via a disulfide bond. Commercial preparations associate this active protein with neurotoxin-associated complexing proteins (NAPs) that maintain stability [16]. Shelf-stable unreconstituted BoNT generally comes as a powder in vials containing 50–500 units, and can be diluted to varying concentrations with 0.9% sodium chloride in preservative-free saline (NaCl/H_2_O) solution [17]. This reconstitution raises the pH, which promotes dissociation of the free, active toxin from the NAPs. This higher pH is also maintained in the target muscle tissue, where the reconstituted BoNT is directly injected for all clinical indications for muscle hypertonicity.

BoNT type A (BoNT-A) enters the presynaptic nerve cell when the heavy chain of the active toxin binds to gangliosides, and then to glycoprotein 2 (also known as SV2) found on and in neuron membranes, resulting in the anchoring of the toxin to vesicles of acetylcholine (ACh) during endocytosis into the presynaptic neuron. BoNT type B (BoNT-B) and other serotypes enter the nerve cell via binding to synaptotagmin receptors on the nerve cell membrane [16]. Once inside the neuron, BoNT impairs the exocytosis of vesicles by interfering with the normal function of the soluble N-ethylmaleimide-sensitive factor attachment protein receptor (SNARE) complex that normally brings these vesicles to the outer membrane at the motor endplate of the neuromuscular junction, where they can be released. BoNT-A exerts this effect on the SNARE complex by cleaving synaptosomal-associated proteins of 25kDa (SNAP-25), while BoNT-B cleaves synaptobrevin, also known as vesicle-associated membrane protein (VAMP) [18].

Many of those treated with BoNT report improvement in their dystonia without significant weakness, or maintain improvement in symptoms after the recovery of weakness [19]. Given this dissociation, other effects of BoNT are hypothesized to contribute to its clinical benefits. BoNT is used clinically in small enough amounts to limit its direct effects on the central nervous system, but it may have indirect effects on the cortex and spinal cord via afferent pathways from muscle fibers. For example, methods such as electromyography (EMG) and transcranial magnetic stimulation (TMS) have been used to show an association with BoNT use in focal dystonia, and a reduction in reflexes and evoked potentials associated with cortical activity and plasticity [20,21]. In one TMS study, the motor-evoked potential was reduced after BoNT, but this effect was attenuated over time after the BoNT treatment and was correlated with a clinical dystonia score [21]. Dystonic pain may also be relieved, not only through the decrease in the contraction of overactive muscles, but through BoNT’s blockade of other substances associated with pain, such as substance P, glutamate, and calcitonin gene-related peptide [18].

## 4. Approved Indications for BoNT

Various formulations of BoNT are clinically available in the United States, including onabotulinumtoxinA (Ona-BoNT-A, name-brand Botox^®^), daxibotulinumtoxinA-lanm (Daxi-BoNT-A, name-brand Daxxify^®^), abobotulinumtoxinA (Abo-BoNT-A, name-brand Dysport^®^), prabotulinumtoxinA-xvfs (Praba-BoNT-A, name-brand Jeuveau^®^), letibotulinumtoxinA-wlbg (Leti-BoNT-A, name-brand Letybo^®^), rimabotulinumtoxinB (Rima-BoNT-B, name-brand Myobloc^®^), and incobotulinumtoxinA (Inco-BoNT-A, name-brand Xeomin^®^) [22,23,24,25,26,27,28]. Cosmetic and therapeutic indications approved by the Food and Drug Administration (FDA) for each toxin can vary. FDA-approved indications for various formulations include chronic migraine, urinary conditions such as overactive bladder, axillary hyperhidrosis, strabismus, blepharospasm, sialorrhea, and the appearance of facial lines. All of these formulations are approved for cervical dystonia, except Praba-BoNT-A and Leti-BoNT-A (approved for the cosmetic appearance of facial lines only), and all of the other BoNT-A toxins, except Daxi-BoNT-A, are approved for spasticity (only in the upper limb in the case of Inco-BoNT-A). In the United States, FDA guidelines recommend a maximum of 400 units of Ona-BoNT-A in a 3-month interval, and many insurers will approve up to 600 units during that time, which limits its use in generalized dystonia and widespread spasticity, which would theoretically require larger doses to treat every involved muscle and would be more impractical to use.

Despite the widespread use and FDA approval for spasticity, which may be useful within dosage limitations for post-stroke hemiparetic spasticity, none of the formulations are FDA-approved for FLD, even though the focality of this diagnosis makes it an even more ideal candidate than multi-limb spasticity for targeted treatment within dosing limitation guidelines. There is a high prevalence of post-stroke spasticity. Given an estimated 25.3% of stroke patients affected by spasticity and an estimated 3.3% prevalence of stroke within the United States, this would translate to roughly 830 per 100,000 [29,30]. By comparison, primary dystonias have an estimated prevalence of 16.43 per 100,000 [31]. This lower prevalence impacts the feasibility of clinical trial accrual, especially larger phase III trials required for FDA approval.

However, there are FDA approvals for some focal dystonias, such as cervical dystonia and blepharospasm, which have estimated worldwide prevalences of 5 per 100,000 and 4.2 per 100,000, respectively, in one meta-analysis of primary dystonias [31]. After this, task-specific dystonias, such as musician’s dystonia and writer’s cramp, are some of the most common dystonias. BoNT is frequently used in clinical practice and research studies for these indications, despite the lack of FDA approval, and, in our experience, may still be covered as off-label indications by some insurers.

## 5. Dosing

The dosing of BoNT for dystonia and other indications in which the target muscles are still functional is generally lower than for spasticity [17,32]. More diluted solutions of reconstituted BoNT inherently involve larger volumes, which may result in greater spread to adjacent structures [33]. This is more of a problem for smaller muscles of the upper extremity, often affected in task-specific dystonias, where space is at a premium. Smaller volumes of higher concentrations can be used to reduce discomfort and spread to nearby muscles vital to the accurate performance of the task. This may also affect the choice of the particular BoNT formulation. While dose conversions are not recommended by FDA labeling or the manufacturers, studies indicate that 1 unit of Ona-BoNT-A is generally considered equivalent to 1 unit of Inca-BoNT-A and these formulations are equivalent to somewhere between 2 and 5 units of Abo-BoNT-A, or 50 units of Rima-BoNT-B [17,34]. Given this, more potent formulations may be desired for focal dystonias.

Recommendations for specific dilutions and dosages of BoNT are not specifically targeted to FLDs, and are described primarily in other indications, such as spasticity, cervical dystonia, or facial dystonias/cosmetic use [33,35,36]. One consensus guideline suggested a standard dilution for Ona-BoNT-A and Inco-BoNT-A of 2.5 mL NaCl/H_2_O per 100 units regardless of the indication, resulting in individual volumes of 0.5 mL (20 units) for injection sites outside the face, with doses able to be adjusted based on the baseline functionality and mass of the muscle, as well as the demographic factors of the target patient [17]. FDA recommendations for cervical dystonia include an alternative suggested dilution of 2mL NaCl per 100 units of Ona-BoNT-A, and this dilution is often used clinically for other forms of dystonia despite a lack of consensus guidelines on dosing in dystonia [22].

A general review of dosing for BoNT in a variety of indications, including various types of dystonia, included a table denoting specific numeric ranges of units for all indications except FLD and dystonic tics [37]. For these indications, the recommendation was “individualized doses,” though a descriptive recommendation for writer’s cramp elsewhere in the article suggested 5 units for smaller hand muscles and 10–20 units for larger forearm muscles. Guidelines for specific dilutions or particular BoNT formulations were not provided. One review indicated doses in focal hand dystonia of 25–50 units of Ona-BoNT-A and 112–127 units of Abo-BoNT-A [32].

Our clinical trial in musician’s dystonia used individualized dosing for each subject, averaging about 22.8 units of Inca-BoNT-A at the first visit, with some subjects given boosters 2 and/or 4 weeks later [3]. Another trial in writer’s cramp used a standard dilution of 20 units of Abo-BoNT-A per 0.1 mL NaCl/H_2_O, resulting in injections as small as 10–15 units (0.05–0.075 mL) per fascicle in finger extensors and up to 60 units (0.3 mL) per fascicle in finger flexors, with no follow-up injections included in the analysis [38].

## 6. Muscle Selection and Targeting

In addition to the consideration of BoNT dosing limitations, muscle selection must be based on careful history and the examination of involved muscles, as described by Karp and Alter in their review of muscle selection for FLD [32]. This may include the patient’s own report of activities that induce symptoms, areas of discomfort, and the description of abnormal movements. The physical examination involves the assessment of the muscles at rest and with provoking activities in the context of knowledge about the muscles involved in normal movement for a given activity. For example, to assess writer’s cramp, one must understand the usual finger flexion involved in writing, the differences in wrist flexion that may occur between left- and right-handed writers, and the more unique postures specific to the individual. Assessment of gait and wear patterns on shoes may be helpful in lower extremity dystonias. It may be difficult to distinguish dystonic muscles from those that are activated as a form of compensation to oppose the dystonia or otherwise allow for somewhat functional movement, as the activation of either dystonic muscles or compensatory muscles may cause discomfort to the patient.

Targeting areas of the muscle with high amounts of neural arborization may allow smaller doses of BoNT to be used to provide the same effect as larger, untargeted injections, since injections would be closer to areas with a higher concentration of motor endplates. One research group has used novel staining techniques in cadaver muscle to demonstrate these highly concentrated areas in a number of larger individual muscles of the upper extremity, including the triceps, supraspinatus, and rhomboid muscles [39,40,41]. In their study of the triceps muscle, sections of the muscle between the acromion and olecranon were divided in tenths, and particular tenths were highlighted as optimal for injection based on high levels of arborization [39]. Although this technique may be helpful for targeting muscles that have been studied using this staining technique and other similar methodologies, not every muscle that might be targeted for BoNT use has been studied in this way. The data cannot necessarily be extrapolated for use in other muscles, and targeting in this way would be difficult for very small muscles, such as finger flexors involved in task-specific dystonias. Even in larger muscles, where localization using bony and muscular landmarks is possible, visual inspection alone may not be sufficiently accurate for the correct targeting of muscles, as evidenced by one trial in which only about 38% of targeted flexor or extensor muscles in focal hand dystonia were correctly localized using surface anatomy when confirmed by EMG [42].

EMG may be used for localization by having the patient purposefully perform the motion known to activate the target muscle while the EMG needle is inserted. For example, if a patient flexes his or her wrist while the EMG needle is correctly inserted into a wrist flexor, voluntary motor unit action potentials will be heard and visualized. Specialized, hollow EMG injection needles can be used that allow BoNT to be injected immediately after localization in this manner. If the needle is inserted in a muscle that performs a different action than that being targeted, these action potentials will be absent, and the needle can be moved until the correct muscle is found.

In electrical stimulation (e-stim), a similar needle is inserted and used as a focal point for stimulation that induces muscle contractions without requiring voluntary motion on the part of the patient. For example, if a wrist flexor is being targeted, e-stim of the correct muscle with the injection needle will induce visible wrist flexion as confirmation of correct localization. Ultrasound may be used to visualize the location of the inserted needle and the target muscle using ultrasound landmarks, though using this technique alone relies on a greater degree of ultrasound knowledge. Ultrasound alone can also be limited in cases when target muscles are small, poorly delineated, and/or grouped closely together, such as the flexors for individual fingers and toes.

Larger reviews have suggested that guidance using ultrasound, EMG, and e-stim are superior to use of anatomical landmarks in spasticity and cervical dystonia, but such studies did not indicate one method as superior to another [43,44]. One randomized-controlled study of 12 subjects with writer’s cramp suggested improved strength and subjective benefit with e-stim compared to EMG guidance for BoNT injection, but this study was limited by poor blinding, as one of the two raters of muscle strength for each subject was aware of the guidance technique used [45].

In our experience, ultrasound may be used in combination with e-stim for the initial targeting and confirmation of placement. Ultrasound first allows visual guidance before inserting the needle. Once inserted, small contractions of the target muscle in response to e-stim can be visualized on ultrasound at lower, less noxious e-stim levels. After this initial confirmation, the e-stim level can be increased until visible target movement is seen on the patient’s limb, thus confirming the correct placement. Correct targeting will maximize the beneficial effect of the BoNT treatment and make it less likely that doses are inappropriately increased during treatment adjustment in subsequent visits based solely on the lack of therapeutic effect, which might better be explained by incorrect targeting.

## 7. Complications of Treatment

The main side effect of treatment is weakness in the targeted muscles, or potentially in adjacent muscles and nerves via the local spread of BoNT or inaccurately targeted muscles. Weakness in the target muscles for facial dystonias may have primarily cosmetic or communicative implications, and weakness in the setting of BoNT use for spasticity may affect muscles that already have limited functionality at baseline [17]. However, weakness in dystonic limbs may cause more functional impairment.

Task-specific dystonias are generally more distal [1,17], so while weakness in targeted muscles may be less likely to completely prevent ambulation or large movements, they can have a larger effect on coordination and dexterity. For example, maintenance of distal finger flexion is a vital component of stability and instrument activation for musicians, which limits how much the flexor digitorum profundus can be targeted in musician’s dystonia [3]. Systemic effects of BoNT are rare, but anticholinergic effects may occur, particularly with BoNT-B [46].

Another factor that can affect the treatment efficacy is the decreased or completely lost response in some patients to subsequent injections of BoNT. Some of this has been attributed to immunogenicity and the development of antibodies to proteins in BoNT. Antibody levels have been associated with shorter intervals between doses, higher doses, and certain formulations of BoNT, such as Rima-BoNT-B [34]. This effect can be countered with longer intervals between dosing, the initial use of formulations less associated with immunogenicity, like Inco-BoNT-A, and switching to alternate formulations. In our study of BoNT-A in musicians’ dystonia, using booster injections, no neutralizing antibody formation was found [3].

## 8. Assessment of Benefit

Overall, clinical trials investigating FLDs have tended to have small sample sizes, which is reflective of the rarity of this pathology, but they have shown benefit in musician’s dystonia, dystonic tremor, writer’s cramp, and generally in focal hand dystonia of all causes [3,38,47,48,49,50,51], as noted in Table 1 and Table 2 regarding trials and reviews, respectively. There are no controlled trials dedicated to focal leg dystonia, such as runner’s dystonia. One trial of 17 patients included 1 patient with leg dystonia in the setting of parkinsonism and 2 patients with idiopathic leg dystonias, but this trial did not distinguish between arm and leg dystonias in the analysis, nor did it show a statistically significant benefit from BoNT compared to the placebo in the subject group as a whole [50]. One review of clinical trials specific to writer’s cramp and musician’s dystonia counted only 139 patients across all trials [52].

Clinical trials of BoNT in FLD have used a variety of primary and secondary outcome measures to assess subject-reported and investigator-evaluated changes in movement and quality of life. Subjects’ self-ratings are frequently assessed using a Visual Analog Scale (VAS), in which they can report their functional movement, such as musical performance or handwriting in the case of some task-specific dystonias, by marking on a line scale from 0 (worst) to an upper limit set by each study, such as 10 or 100 (best performance/function). Other self-reported scales of the subject’s functional status, symptom severity, and level of disability are also used.

Investigator evaluation may involve the grading of dystonia improvement, such as one evaluation of musician’s dystonia using a seven-point clinical global impression (CGI) scale, where the level of dystonia after the treatment could be rated on a scale from −3 (marked worsening) to +3 (marked improvement) [3]. In writer’s cramp, the writing speed may be assessed. Weakness can be measured using the Medical Research Council (MRC) strength grading from zero (no movement) to five (full strength). This scale is often used outside of research in the clinical setting to evaluate muscle movement in various planes visually and against the evaluator’s opposing movements. Dynamometry can also be used to measure the force of the muscle contraction of the affected parts of the upper limb [3].

## 9. Conclusions

The relatively low prevalence of FLDs and a relative paucity of controlled trials have limited the FDA approval of BoNT for this indication due to difficulties with the sufficient recruitment of subjects for high-powered clinical trials. Small controlled clinical trials have demonstrated its efficacy, particularly for focal hand dystonias, such as the task-specific writer’s cramp and musician’s dystonia, which can be career-ending when not adequately treated. However, it is difficult to compare the efficacy among these trials, as each trial measured the efficacy of BoNT by using a unique combination of subjective and objective outcome measures from the subject and outside observers. These markers of efficacy included a numerical rating of the task performance and dystonia severity, the subject’s desire to continue treatment, and other quality of life concerns such as pain, depression, and disability. The importance of each of these factors to each patient will vary, although, in general, measures of “performance” will tend to be more important in task-specific dystonias, and these measures were significantly improved in those trials that included these measures. Injection techniques and dosages have also not been identical across trials, which may be related to the difficulty in creating targeted dosage guidelines for a pathology that often presents so uniquely for different patients, particularly in terms of which muscles are involved and the degree of involvement of each muscle. Although the choice of muscles to inject for a given case of dystonia should be individualized, large, multicenter trials would need to find a way to standardize the dosages to use for particular muscles. A unified injection and guidance technique would be needed despite the inclusion of multiple investigators, who may use different methods in their own clinical practices.

Despite these limitations, BoNT is often used in clinical practice for focal dystonias. BoNT allows for local treatment that can be accurately targeted using imaging and electrophysiological methods in order to provide effects in desired muscles with limited local and very rare systemic side effects, all of which are temporary and reversible. Increased awareness of these diagnoses of dystonia and training in BoNT for their treatment will allow more clinicians a greater level of familiarity and comfort to plan and administer this targeted treatment. Greater awareness and access may also facilitate the research needed to gain FDA approval in the future, which may require more trials with increased standardization, greater inclusion of leg dystonias, validated outcome measures, and quantitative meta-analyses of existing trials.

## Figures and Tables

**Table 1 toxins-17-00122-t001:** Clinical trials assessing the benefit of BoNT in FLDs.

First Author & Year of Publication	Design of the Studies	Characteristics of Patients	Assessment	Therapy	Treated Muscles	Treatment Session	Follow Up	Outcomes	Conclusions
**Frucht, S.J. 2024 [3]**	Double-blind, placebo-controlled, cross-over study	21 professional musicians with focal upper extremity task-specific dystonia	Assessment by CONSORT criteria	Group A: 13 patients naïve to BoNT Group B: 8 patients with prior BoNT treatment. 11 patients were randomized to placebo in cycle 1 and active drug in cycle 2 (*P*→*A*); 10 patients were randomized to receive active drug in cycle 1 and then placebo in cycle 2 (*A*→*P*).	Upper limb muscles	The mean dose of Inco-BoNT-A injected at first dose was 22.8 U (7.5–45.0 U). Nine patients received boosters at week 2 of the active arm (mean dose 16.7, range 5–35 U), and ten patients received boosters at week 4 of the active arm (mean dose 8.5 U, range 0.0–30.0)	The primary outcome measure for cycle 1 week 8 in comparison to baseline	Analysis of the primary outcome measure in comparison to baseline revealed a change in dystonia severity of *p* = 0.04 and an improvement in overall musical performance of *p* = 0.027.	Statistically significant efficacy of Inco-BoNT-A injections in FTSDma.
**Kruisdijk JJM** **2006 [38]**	Double-blind, randomized, placebo-controlled trial and to evaluate the follow-up results	39 patients with signs and symptoms of idiopathic writer’s cramp	Assessments were made at baseline and 2 months (secondary outcome) and 3 months (primary outcome). Duration of follow-up was 1 year.	Trial treatment consisted of BoNT-A injections or placebo injections.	Finger flexor muscles, finger extensors, wrist flexors, wrist extensors	Muscles were selected for injection according to the pattern of movements and visible or palpable hypertonia. Finger flexor muscles were injected with 60 IU (0.3 mL) per fascicle, finger extensors with 10–15 IU (0.05–0.075 mL) per fascicle, wrist flexors with 60–100 IU (0.3–0.5 mL) and wrist extensors with 30–40 IU (0.15–0.2 mL).	1 year	The primary outcome measure demonstrated the benefit of the treatment and above all the patients’ request to continue the treatment, despite its possible disadvantages.	Treatment with BoNT-A injections led to a significantly greater improvement compared with placebo, according to patients’ opinion and clinical assessment scales.
**Tsui, J.K.C.** **1993 [51]**	Double-blind, placebo-controlled study	20 patients with writer’s cramp. 13 men and 7 women, with a mean age of 41.75 years (range 28 to 58 years)	Before each treatment and 2 and 6 weeks after each treatment	BoNT treatment or placebo	Flexor digitorum superficialis (30 MU), flexor digitorum profundus (30 MU), flexor carpi radialis (30 MU), flexor carpi ulnaris (30 MU), flexor pollicis longus (25 MU), extensor digitorum (50 MU), extensor carpi ulnaris (30 MU), pronator teres (30 MU), and pronator quadratus (30 MU)	Each patient was given 2 sets of injection, with a treatment interval of 3 months	2 and 6 weeks after each treatment	In the BoNT-A group 8 of the 12 patients reported pain relief, 4 patients reported definite improvement in writing (score of 2), 2 patients reported slight subjective improvement in writing (score of 1). No changes in writing were noted in 13 patients (score of 0). In one patient, pen control was worse (score of −1) for 8 days after treatment (weakness)	BoNT-A injections are effective in relieving symptoms only in selected cases of writer’s cramp
**Yoshimura, D.M** **1992 [50]**	Placebo-controlled, blinded study.	17 patients with limb dystonias (10 with occupational cramps, 3 with idiopathic dystonia unrelated to activity, and 2 each with post-stroke and parkinsonian dystonia)	Assessment was performed by review of the randomly recorded videotape, independently by three physicians in a blinded manner. Only severity was scored	BoNT treatment or placebo	Muscles were identified both clinically and by EMG on the basis of the abnormal posture	Each patient received a series of four injections into the affected muscles via EMG guidance. 3 BoNT-A treatment and 1 placebo treatment	1 month after treatment	In 82% of patients, subjective improvement was seen with at least one dose of BoNT Objective benefit was seen in 59% of patients with at least one dose of BoNT	BoNT-A provides subjective benefit, and it is a safe therapy for limb dystonia.

CONSORT = The Consolidated Standards of Reporting Trials; FTSDma = focal task-specific dystonia of musician’s arm; U = units; MU = mouse units.

**Table 2 toxins-17-00122-t002:** Meta-analysis and reviews assessing the benefit of BoNT in FLDs.

First Author and Year of Publication	Design of the Studies	Characteristics of Studies	Assessment	Therapy	Treated Muscles	Outcomes	Conclusions
**Del Sorbo, F. 2015 [47]**	To review the primary and secondary outcome measures implemented by recently reviewed studies To verify whether they corresponded with the recommendations set forth by a rating scales task force	Meta-analysis included a total of 42 clinical trials (20 placebo-controlled, 10 active comparator, and 12 methodological or uncontrolled studies)	Assessments by CONSORT criteria Outcome measures included tools related to motor and non-motor features, such as pain and depression, and functional as well as health- related quality of life features	BoNT treatment	Blepharospasm, oromandibular, laryngeal, cervical, and upper limb dystonias	Recommended rating scales were only used in studies for cervical dystonia and blepharospasm • For cervical dystonia alone, there was a robust fitting of a recommended scale (the TWSTRS scale) with a placebo-controlled study design • The evidence collected on upper limb and laryngeal dystonias was based on the use of non-recommended outcome measures	Implementation of recommended rating instruments by high-quality studies would provide the strongest possible evidence of the efficacy of BoNTs in focal dystonias; by contrast, the implementation of non-recommended rating tools would weaken even studies with a methodologically strong design
**Zakin, E. 2021 [52]**	Review	139 patients across the six selected studies, with 99 individuals affected by writer’s cramp and 40 individuals with other task-specific focal dystonias, inclusive of musician’s dystonia; the age range included was 20–80 years old	Various severity scales to quantify the response to toxin injection, with ratings of instrument or pen control included as subjective ratings	BoNT-A treatment	Forearm muscles (either flexors or extensors)	Pooled data for toxin response showed that 73% of patients who received the drug demonstrated improvement	BoNT-A injections should be considered as an integral therapeutic option for the management of focal task-specific dystonia, especially in individuals who have demonstrated lack of benefit or adverse side effects with oral pharmacological agents

TWSTRS = Toronto Western Spasmodic Torticollis Rating Scale.

## Data Availability

No new data were created or analyzed in this study.

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
