# Peer review of "Review: Botulinum Toxin for Treatment of Focal Limb Dystonia"

_toxins, 2025, doi:10.3390/toxins17030122_

Round 1
Reviewer 1 Report
Comments and Suggestions for Authors
The manuscript presents a comprehensive review of botulinum toxin (BoNT) for focal limb dystonia (FLD). The discussion is well-structured, covering epidemiology, pathophysiology, clinical application, dosing strategies, and challenges in BoNT treatment. The authors have effectively summarized relevant clinical studies and the current state of knowledge regarding the off-label use of BoNT for FLD. However, several areas require clarification, additional citations, and refinement to enhance scientific rigor.
Major Comments:
-
Citations for Innervation and Muscle Targeting in BoNT Injections
The discussion on muscle selection and targeting for BoNT injections lacks sufficient anatomical references regarding intramuscular innervation patterns. Given that precise localization is crucial for optimizing efficacy and minimizing side effects, I strongly recommend that the authors cite the following study:- Yi KH, Lee JH, Hur HW, Lee HJ, Choi YJ, Kim HJ. Distribution of the intramuscular innervation of the triceps brachii: Clinical importance in the treatment of spasticity with botulinum neurotoxin. Clin Anat. 2023 Oct;36(7):964-970. doi: 10.1002/ca.24004.
This paper provides clinically relevant anatomical data on the intramuscular innervation of the triceps brachii, which is essential in planning effective BoNT injections for spasticity and dystonia. Its inclusion would strengthen the manuscript’s discussion on optimizing injection techniques.
-
Lack of High-Powered Clinical Trials and FDA Approval Challenges
The manuscript attributes the lack of FDA approval for FLD treatment to low disease prevalence. While this is a valid concern, the authors should also discuss challenges in conducting large-scale, high-powered randomized controlled trials (RCTs) for BoNT, including variability in injection techniques, individual responses, and dose standardization. -
Role of Imaging and Electrophysiological Guidance in BoNT Targeting
The manuscript discusses the limitations of anatomical landmark-based injections but does not provide enough detail on how ultrasound and electromyography (EMG) guidance can improve precision. A brief discussion on the comparative benefits of these techniques would be beneficial.
Minor Comments:
- Terminology Consistency: Ensure consistency in terminology throughout the manuscript, particularly in the use of “BoNT” vs. “botulinum toxin.”
- Table 1 Formatting: Some entries in Table 1 lack clear separation between the study design and results, making it difficult to follow. Please revise for clarity.
- Abbreviations: Define all abbreviations upon first use to enhance readability.
Conclusion:
This is a well-written and informative review on the use of BoNT for FLD. Incorporating the suggested citation and addressing the above points will improve the manuscript’s clarity, depth, and scientific robustness.
Recommendation: Minor revisions required before acceptance.
Comments on the Quality of English LanguageN/A
Author Response
Major Comments:
Comments 1: Citations for Innervation and Muscle Targeting in BoNT Injections
The discussion on muscle selection and targeting for BoNT injections lacks sufficient anatomical references regarding intramuscular innervation patterns. Given that precise localization is crucial for optimizing efficacy and minimizing side effects, I strongly recommend that the authors cite the following study:
- Yi KH, Lee JH, Hur HW, Lee HJ, Choi YJ, Kim HJ. Distribution of the intramuscular innervation of the triceps brachii: Clinical importance in the treatment of spasticity with botulinum neurotoxin. Clin Anat. 2023 Oct;36(7):964-970. doi: 10.1002/ca.24004.
This paper provides clinically relevant anatomical data on the intramuscular innervation of the triceps brachii, which is essential in planning effective BoNT injections for spasticity and dystonia. Its inclusion would strengthen the manuscript’s discussion on optimizing injection techniques.
Response 1: Thank you for this citation. After reviewing the requested article and choice additional articles cited therein, we have included additions in Section 6 (Muscle Selection and Targeting), paragraph 2 on page 5 of the tracked changes document, which summarize the Yi article’s methods, implications in targeting, and limitations in the setting of other techniques.
Comments 2: Lack of High-Powered Clinical Trials and FDA Approval Challenges
The manuscript attributes the lack of FDA approval for FLD treatment to low disease prevalence. While this is a valid concern, the authors should also discuss challenges in conducting large-scale, high-powered randomized controlled trials (RCTs) for BoNT, including variability in injection techniques, individual responses, and dose standardization.
Response 2: Thank you for the feedback. We have edited Section 9 (Conclusions) on page 13 and 14 of the tracked changes document to discuss differences in outcome measures, techniques, and doses in existing trials, as well as to highlight the difficulties of standardization when applied to a larger trial and propose which aspects should and should not be standardized.
Comments 3: Role of Imaging and Electrophysiological Guidance in BoNT Targeting
The manuscript discusses the limitations of anatomical landmark-based injections but does not provide enough detail on how ultrasound and electromyography (EMG) guidance can improve precision. A brief discussion on the comparative benefits of these techniques would be beneficial.
Response 3: Thank you for this suggestion. Additions and clarifications were added throughout Section 6 (Muscle Selection and Targeting), creating updated paragraphs 3-6 on page 6 of the tracked changes document. Further description and examples are given to demonstrate how EMG, electrical stimulation (e-stim), and ultrasound improve precision alone and together, including additional references comparing guidance techniques.
Minor Comments:
Comments 4: Terminology Consistency: Ensure consistency in terminology throughout the manuscript, particularly in the use of “BoNT” vs. “botulinum toxin.”
Response 4: Thank you for pointing this out. We have made adjustments to substitute longer names for subtypes of botulinum toxin for abbreviations using BoNT in Section 4 (Approved Indications for BoNT), lines 116-121 in paragraph 1 on page 3 of the tracked changes document, and later references to these subtypes have been adjusted throughout Section 5 (Dosing) on pages 4-5 of the tracked changes document, Section 7 (Complications of Treatment) on page 7 of the tracked changes document, and in the cell for the “Yoshimura” row, “Outcomes” column of Table 1 on page 10 of the tracked changes document. The title of Table 1 was also edited to replace “Focal Limb Dystonias” with FLDs on page 7, line 300 of the tracked changes document, and the same abbreviation was used in the new Table 2 on page 11, line 306 of the tracked changes document. After review of the document, no other specific terminology inconsistencies were noted.
Comments 5: Table 1 Formatting: Some entries in Table 1 lack clear separation between the study design and results, making it difficult to follow. Please revise for clarity.
Response 5: Thank you for the feedback. The tables were separated into “Table 1. Clinical trials assessing benefit of BoNT in FLDs” and “Table 2. Meta-analysis and reviews assessing benefit of BoNT in FLDs” just below Table 1. The “Del Sorbo” and “Zakin” rows were moved from Table 1 to 2, and the “Treatment Session” and “Follow up” columns are not used in Table 2. Within Table 1, the cell for the “Kruisdijk” row, “Outcomes” column on page 8 of the tracked changes document is edited to clarify the results. The cells for the “Yoshimura” row, “Outcomes” and “Conclusions” columns on page 10 of the tracked changes document are also edited for clarity.
Changes were also made within the new Table 2 on pages 11-12 of the tracked changes document. In the cell for the “Zakin” row, “Therapy” column on page 12 of the tracked changes document, the text “Studies using only the BoNT-A serotype were included” was replaced with “BoNT-A treatments” to better fit the wording of the other cells in this column. Significant edits were also made to the “Del Sorbo” row on pages 11-12, with text added, removed, and moved amongst columns to clarify design and results. The tracked changes cannot be seen as easily due to the limitations of the word processing program with recognizing changes to text that has been cut and pasted. These changes are described in more detail below:
- The text from the “Design of the studies” column reading “Meta-analysis included a total of 42 clinical trials (twenty placebo-controlled, ten active comparator, and twelve methodological or uncontrolled studies)” was moved to the “Characteristics of studies” column, with addition of numerical abbreviations to reduce crowding within the table cell.
- The prior text in the “Characteristics of studies” column was removed, which previously read “10 studies dealt with blepharospasm, two with oromandibular dystonia, thirteen with cervical dystonia, six with limb dystonia, and nine with laryngeal dystonia.”
- New text was added to the “Design of the studies” column, now reading “To review the primary and secondary outcome measures implemented by recently reviewed studies” and “To verify whether they corresponded with recommendations set forth by a rating scales task force.”
- The text within the “Outcomes” column reading “Outcome measures included tools related to motor and non-motor features, such as pain and depression, and functional as well as health- related quality of life features” was moved to the “Assessment” column.
- The text within the “Conclusions” column, reading “Recommended rating scales were only used in studies for cervical dystonia and blepharospasm. For cervical dystonia alone there was a robust fitting of a recommended scale (the TWSTRS scale) with a placebo-controlled study design. The evidence collected on upper limb and laryngeal dystonia was based on the use of non-recommended outcome measures” was moved to the “Outcomes” column.
- Additional text was added to the “Conclusions” column, reading “Implementation of recommended rating instruments by high quality studies would provide the strongest possible evidence of efficacy of BoNTs in focal dystonias. By contrast, implementation of non-recommended rating tools would weaken even studies with a methodologically strong design.”
Comments 6: Abbreviations: Define all abbreviations upon first use to enhance readability.
Response 6: Thank you for pointing this out. The document was reviewed for abbreviations. Focal limb dystonias (FLDs), botulinum toxin (BoNT), and Food and Drug Administration (FDA) are defined separately in the front matter of the article (Abstract, Keywords, and Key Contribution statement). In the body of the text, these terms are redefined:
- FLD is defined in Section 1 (Defining Focal Limb Dystonias) paragraph 1, line 25 on page 1 of the tracked changes document.
- BoNT is defined in the title of Section 2 (BoNT in the Context of Dystonia Treatment) line 46 on page 2 of the tracked changes document.
- Neurotoxin-associated complexing proteins (NAPs) and 0.9% sodium chloride in preservative-free saline (NaCl/H2O) solution are defined on line 79 and 82 respectively in Section 3 (Physiology), paragraph 2 on page 2 of the tracked changes document.
- BoNT type A and type B (BoNT-A and BoNT-B) are defined in Section 3 (Physiology) on line 86 and line 89 respectively in paragraph 3 on page 3 of the tracked changes document.
- Acetylcholine (ACh), N-ethylmaleimide-sensitive factor attachment protein receptor (SNARE), synaptosomal-associated proteins of 25kDa (SNAP-25), and vesicle-associated membrane protein (VAMP) are defined in Section 3 (Physiology) on lines 87, 89, 94, and 96 respectively in paragraph 3 on page 3 of the tracked changes document.
- Electromyography (EMG) and transcranial magnetic stimulation (TMS) are defined in Section 3 (Physiology) in paragraph 4, line 106 on page 3 of the tracked changes document.
- Subtypes of BoNT are defined in Section 4 (Approved Indications for BoNT), throughout paragraph 1 on page 3 as delineated in Response 4 above.
- FDA is defined in Section 4, paragraph 1, line 121 of page 3 of the tracked changes document.
- Electrical stimulation (e-stim) is defined in Section 6 (Muscle Selection and Targeting), paragraph 4, line 238 on page 6 of the tracked changes document.
- The Consolidated Standards of Reporting Trials (CONSORT), focal task-specific dystonia of musician’s arm (FTSDma), Units (U) and mouse units (MU) are all abbreviations used in Table 1 and defined directly under the table in Section 8 (Assessment of Benefit) on lines 301-304 on page 11 of the tracked changes document.
- Toronto Western Spasmodic Torticollis Rating Scale (TWSTRS) used in Table 2 is defined directly under the table in Section 8 (Assessment of Benefit) on line 307 on page 12 of the tracked changes document.
- Visual Analogue Scale (VAS) is defined on line 307, clinical global impression (CGI) is defined on line 313, and Medical Research Council (MRC) is defined on line 316 of Section 8 (Assessment of Benefit), page 11 below the table of the tracked changes document.
Response to Comments on the Quality of English Language
Point 1: The tickbox for “The English could be improved to more clearly express the research” is selected for the “Quality of English Language” question, and “Comments on the Quality of English Language” is filled by the reviewer as “N/A.”
Response 1: Should the selected tickbox calling for English language improvement be referring to the points described in the “Minor Comments” section of the review, these points have been addressed above in Responses 4-6. If there are other concerns about English language use that need to be addressed, we would appreciate further guidance.
Reviewer 2 Report
Comments and Suggestions for Authors
The review deals with a potentially important topic, as it presents the current knowledge on the subject of BTX use in the treatment of focal limb dystonia. The paper is interesting, concise, and well-written. However, it should be stressed, that the use of BTX as the treatment of focal dystonia has been widely described by other authors, the most important information has been summarized for instance by Zoon et al. in 2012 (Botulinum toxin as treatment for focal dystonia: a systematic review of the pharmaco-therapeutic and pharmaco-economic value; DOI: 10.1007/s00415-012-6510-x). In the study under review, the authors describe the topic in a more detailed fashion, but the existing reviews lower the scientific novelty of the contribution.
Author Response
Comments 1: The review deals with a potentially important topic, as it presents the current knowledge on the subject of BTX use in the treatment of focal limb dystonia. The paper is interesting, concise, and well-written. However, it should be stressed, that the use of BTX as the treatment of focal dystonia has been widely described by other authors, the most important information has been summarized for instance by Zoon et al. in 2012 (Botulinum toxin as treatment for focal dystonia: a systematic review of the pharmaco-therapeutic and pharmaco-economic value; DOI: 10.1007/s00415-012-6510-x). In the study under review, the authors describe the topic in a more detailed fashion, but the existing reviews lower the scientific novelty of the contribution.
Response 1: Thank you for your review. We appreciate your understanding of the background literature that already exists on this topic. We believe that our inclusion primarily of references published after 2012, including two new clinical trials, can be considered an appropriate update on this topic in the last 13 years.
Reviewer 3 Report
Comments and Suggestions for Authors
The review considers Botulinum Toxin for Treatment of Focal Limb Dystonia. This is an interesting topic, although I disagree with the authors on the way they present their findings. The table where the review of their search is collected, not only includes clinical analyses but also reviews. I think that they could be separated in order to present the results more precisely.
Furthermore, the summary should reflect something more specific about the studies found, as well as similarities or lack thereof in the results reached.
On the other hand, this is a review, so the authors should reflect in more detail the mechanism of distribution of the toxin. In this way, the different methods used for the administration of the toxin as well as the effectiveness could be followed.
The presentation and design of the review makes it easier for the reader to understand the topic presented.
Author Response
Comments 1: The review considers Botulinum Toxin for Treatment of Focal Limb Dystonia. This is an interesting topic, although I disagree with the authors on the way they present their findings. The table where the review of their search is collected, not only includes clinical analyses but also reviews. I think that they could be separated in order to present the results more precisely.
Response 1: Thank you for the feedback. The tables were separated into “Table 1. Clinical trials assessing benefit of BoNT in FLDs” on pages 7-11 of the tracked changes document, and “Table 2. Meta-analysis and reviews assessing benefit of BoNT in FLDs” on page 11-12 of the tracked changes document, just below Table 1. The “Del Sorbo” and “Zakin” rows were moved to Table 2, and text was edited within both tables for clarity.
Comments 2: Furthermore, the summary should reflect something more specific about the studies found, as well as similarities or lack thereof in the results reached.
Response 2: Thank you for the feedback. We have edited Section 9 (Conclusions) on pages 13 and 14 of the tracked changes document to discuss more specifically some of the differences in the trials that makes direct comparison somewhat difficult, as well as to apply these thoughts to considerations that might be made in larger future trials.
Comments 3: On the other hand, this is a review, so the authors should reflect in more detail the mechanism of distribution of the toxin. In this way, the different methods used for the administration of the toxin as well as the effectiveness could be followed.
Response 3: Thank you for this suggestion. We have edited Section 3 (Physiology) on page 2-3 of the tracked changes document, including new references, to describe the structure of botulinum toxin, its association with additional proteins, active form, and how all of these factors affect its binding and entrance into its site of action at the motor endplate. The motor endplate site of action is then also discussed later in the text in Section 6 (Muscle Selection and Targeting), paragraph 2, lines 211-214 on page 5 of the tracked changes document, where it serves as a point of potential injection site targeting and bridges into a larger new discussion of various injection guidance techniques.
Round 2
Reviewer 1 Report
Comments and Suggestions for Authors
All the correction has been done. no more comments from reviewers.